# Novel Features of Canopy Height Distribution for Aboveground Biomass Estimation Using Machine Learning: A Case Study in Natural Secondary Forests

Ye Ma [1], Lianjun Zhang [2], Jungho Im [3], Yinghui Zhao [1,†] and Zhen Zhen [1,*,†]

[1] Key Laboratory of Sustainable Forest Ecosystem Management-Ministry of Education, School of Forestry, Northeast Forestry University, Harbin 150040, China; maye@nefu.edu.cn (Y.M.); yinghuizhao@nefu.edu.cn (Y.Z.)

[2] Department of Sustainable Resources Management, State University of New York College of Environmental Science and Forestry, Syracuse, NY 13210, USA; lizhang@esf.edu

[3] Department of Urban and Environmental Engineering, Ulsan National Institute of Science and Technology, Ulsan 44610, Republic of Korea; ersgis@unist.ac.kr

* Correspondence: zhzhen@syr.edu

† These authors contributed equally to this work.

**Abstract:** Identifying important factors (e.g., features and prediction models) for forest aboveground biomass (AGB) estimation can provide a vital reference for accurate AGB estimation. This study proposed a novel feature of the canopy height distribution (CHD), a function of canopy height, that is useful for describing canopy structure for AGB estimation of natural secondary forests (NSFs) by fitting a bimodal Gaussian function. Three machine learning models (Support Vector Regression (SVR), Random Forest (RF), and eXtreme Gradient Boosting (Xgboost)) and three deep learning models (One-dimensional Convolutional Neural Network (1D-CNN4), 1D Visual Geometry Group Network (1D-VGG16), and 1D Residual Network (1D-Resnet34)) were applied. A completely randomized design was utilized to investigate the effects of four feature sets (original CHD features, original LiDAR features, the proposed CHD features fitted by the bimodal Gaussian function, and the LiDAR features selected by the recursive feature elimination algorithm) and models on estimating the AGB of NSFs. Results revealed that the models were the most important factor for AGB estimation, followed by the features. The fitted CHD features significantly outperformed the other three feature sets in most cases. When employing the fitted CHD features, the 1D-Renset34 model demonstrates optimal performance ($R^2$ = 0.80, RMSE = 9.58 Mg/ha, rRMSE = 0.09), surpassing not only other deep learning models (e.g.,1D-VGG16: $R^2$ = 0.65, RMSE = 18.55 Mg/ha, rRMSE = 0.17) but also the best machine learning model (RF: $R^2$ = 0.50, RMSE = 19.42 Mg/ha, rRMSE = 0.16). This study highlights the significant role of the new CHD features fitting a bimodal Gaussian function and the effects between the models and the CHD features, which provide the sound foundations for effective estimation of AGB in NSFs.

**Keywords:** AGB; UAV-LiDAR; machine learning models; deep learning models; canopy height distribution; bimodal gaussian function

## 1. Introduction

Northeastern China, along with northeastern North America and Europe, is home to one of the World's three great temperate mixed-species forests [1]. Accurate assessment of the carbon stocks of natural secondary forests (NSFs) in northeastern China is urgent and crucial for the regional carbon cycle, carbon source/sink, climate change mitigation, and sustainable regional development [2], particularly in light of China's recent implementation of a carbon neutrality policy [3]. Given the equivalence of carbon stocks and biomass, forest carbon stocks are typically converted from forest biomass estimation using a biomass

expansion factor (BEF) [4]. Remote sensing inversion has recently been identified as a valuable approach for assessing large-area forest AGB, enabling accurate monitoring at a landscape scale and providing a spatially explicit distribution of forest AGB [5].

Forest AGB can generally be inversed using either an area-based or individual tree-based approach [6]. For the area-based approach (ABA), plot- or stand-level features are extracted from remotely sensed data and applied to establish a global prediction model for forest AGB estimation [7]. For ABA, due to heterogeneous geographical areas, many factors influence the accuracy of AGB estimation based on remotely sensed data, including forest type, data source, and biomass estimation model [8]. During the past two decades, various remote sensing data have been applied for regional and global AGB estimation [9–11]. Among them, Light Detection and Ranging (LiDAR) has become a mainstream tool for characterizing the parameters of complex forests because of its ability to capture fine and accurate three-dimensional forest canopy structure [12,13]. It can also reduce AGB saturation, which often occurs when using passive optical data for biomass estimation [14]. More recently, Unmanned Aerial Vehicle LiDAR (UAV-LiDAR) has a lower flight altitude and a higher point cloud density (around 50 pts/$m^2$ to 350 pts/$m^2$), which provides a reliable data source for fine-scale AGB estimation [15].

What types of input features or attributes are used in forest biomass estimation models substantially affects AGB estimation [16]. Most research extracted canopy height and intensity metrics (e.g., mean, variance, coefficient of variation, and percentiles) from LiDAR point cloud data [17]. Although these metrics have achieved promising results in forest attribute estimation, the selection of the LiDAR metrics is different due to the variety of regions and forest types [15,16,18]. The canopy height distribution (CHD) is the probability of observing the canopy surface at height $h$ [19]. It has great potential for forest AGB estimation as a canopy structural feature that contains numerous pieces of information and has less information loss compared to LiDAR metrics such as height metrics [19,20]. In the subtropical forests of southeast China, CHD features fitted with a Weibull function have been found to produce accurate AGB estimations [20]. In northeastern China, however, the major forest types are mixed-species stands with a more complex stand structure. To our best knowledge, it is uncertain and unclear how effectively the CHD features can be applied to depict the vertical structure of mixed-species forests to estimate their AGB. It is crucial to identify a probability function associated with CHD that can properly describe the vertical structure of the forest canopy. In addition, the number of model input features is an important factor affecting the model's accuracy and efficiency [21]. Thus, feature selection is commonly used to determine the best subset of features, which include Filter (e.g., Pearson) [22], Wrapper (e.g., recursive feature elimination) [23], and Embedded (e.g., L1 regularization) [24]. Previous studies have demonstrated that the accuracy and efficiency of AGB prediction can be significantly enhanced by employing feature selection to identify the optimal subset of features [25].

A prediction model is another important factor in AGB estimation. For restricted data sources and samples, using a model with superior prediction performance is a useful alternative for improving AGB estimation. A plentiful number of non-parametric machine learning (ML) models, such as support vector machines (SVM) and random forests (RF), have been applied to estimate forest AGB. These models have the advantage over multiple linear regression for dealing with complex nonlinear problems and high-dimensional data. However, further improvement is necessary to enhance the accuracy of machine models for AGB estimation in forests with complex conditions [26,27]. Since 2014, deep learning (DL) algorithms have been attracted to the remote sensing community and have been successfully applied in various image classification and regression tasks [28]. Convolutional Neural Networks (CNNs) are a representative of deep learning that is particularly effective and frequently employed in target detection and image recognition [29]. In recent years, the rapid development of some series of CNN models with deep convolutional networks (e.g., Visual Geometry Group Network (VGG [30], Inception [31], Resnet [32])) has given a reliable result for forest AGB estimation. These CNN models have shown outstanding performance

in the fields of image classification and target detection. Using these DL models in the point cloud data to explore their role in forest parameter estimation can have a positive effect on improving the accuracy of forest AGB estimation and achieving more accurate carbon stock estimation.

Given that suitable features are significant for forest AGB estimation in NSFs, this study aims to identify a new LiDAR metric that can describe the vertical structure of forest canopy and estimate forest AGB to replace the traditional metrics (e.g., LiDAR height and intensity metrics) and to identify the important factors affecting forest AGB estimation. The specific objectives of this study are to: (1) develop an efficient function for fitting CHD that is suitable for the AGB estimation of two-story forest stands; (2) explore the performance of four groups of CHD features (i.e., original CHD features, LiDAR metrics, fitted CHD features, and selected LiDAR metrics) and six models (i.e., SVR, RF, Xgboost, 1D-CNN4, 1D-VGG16, and 1D-Resnet34) in forest AGB estimation; and (3) identify the important factors (features, prediction models, and their interactions) affecting forest AGB estimation using a completely randomized design (CRD).

## 2. Materials and Methods

### 2.1. Study Area

The study area is located in the Maoershan Experimental Forest Farm, Shangzhi City, Heilongjiang Province, China (Figure 1). It spans from 127°29′E to 127°44′E and 45°14′N to 45°29′N, covering a total area of 26,048 hectares. The study area has an average altitude of 300 m, with a peak of 805 m. The average annual temperature and precipitation are 2.8 °C and 723 mm, respectively, and the climate is categorized as continental monsoon. The study area is dominated by mixed conifer-deciduous, deciduous, and coniferous forests, with an average canopy closure of about 0.8.

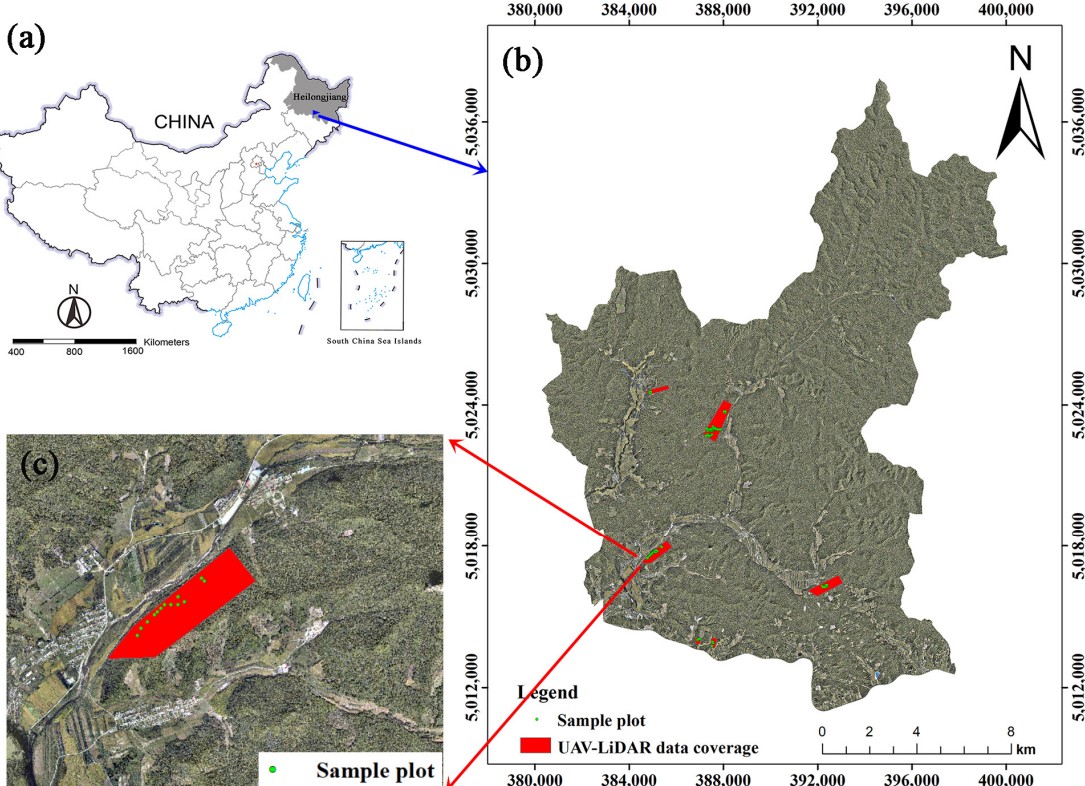

**Figure 1.** (**a**) The location of Heilongjiang Province in China (China Map Examination No. is GS (2019) 1822); (**b**) the location of the study area in Maoershan Forest Farm over a digital Orthophoto map with four UAV flight areas and 62 sample plots; and (**c**) one UAV flight area with 13 sample plots as an example.

## 2.2. Data and Preprocessing

### 2.2.1. Field Inventory Data

A total of 62 mixed conifer-deciduous sample plots (30 × 30 m) were surveyed between 2019 and 2022. The tree species composition of these mixed forests is complex, and a variety of vertical structures exist in the stands, including single-story and two-story forests. The diameter at breast height (DBH, cm), tree height (m), and locations for all trees with the DBH greater than 5 cm were measured using a perimeter ruler, the Vertex IV ultrasound instrument system, and the real-time kinematic (RTK) global navigation satellite system, respectively. Tree species and health conditions were also recorded for all the trees in the plots. Major tree species include Manchurian ash (*Fraxinus mandshurica* Rupr.), Manchurian walnut (*Juglans mandshurica* Maxim.), white birch (*Betula platyphylla* Suk.), Mongolian oak (*Quercus mongolica* Fisch. ex Ledeb.), elm (*Ulmus pumila* L.), Korean aspen (*Populus davidiana*), Miyabe maple (*Acer miyabei* Maxim.), Korean pine (*Pinus koraiensis* Sieb. et Zucc.), and Changbai larch *(Larix olgensis* Henry).

Using species-specific additive biomass equations (see Equation (1)), we calculated the biomass of each tree's components, including stems, branches, and foliage, for northeastern China [1,33]. The AGB of each individual tree was calculated by adding the biomass of each tree's components according to different tree species' parameters, and the stand-level AGB (Mg/ha) was aggregated by all the individual tree AGBs in each plot.

$$lnW_a = ln(W_s + W_b + W_f) = ln(a_s \cdot D^{b_s} + a_b \cdot D^{b_b} + a_f \cdot D^{b_f}) + \varepsilon_a \tag{1}$$

where $W_a$ represents AGB, $W_i$ is biomass (kg) of different components; $D$ is *DBH* (cm); *ln* is the natural logarithm; $\varepsilon_a$ is the error term; $a_i$ and $b_i$ are regression coefficients, $i = [s, b, f]$ represents stem, branch, and foliage, respectively. The $a_i$ and $b_i$ for all the tree species are listed in Table A1. The descriptive statistics of individual tree Height and DBH for the 62 plots are presented in Figure 2. The statistical information of the sample plots is shown in Table 1.

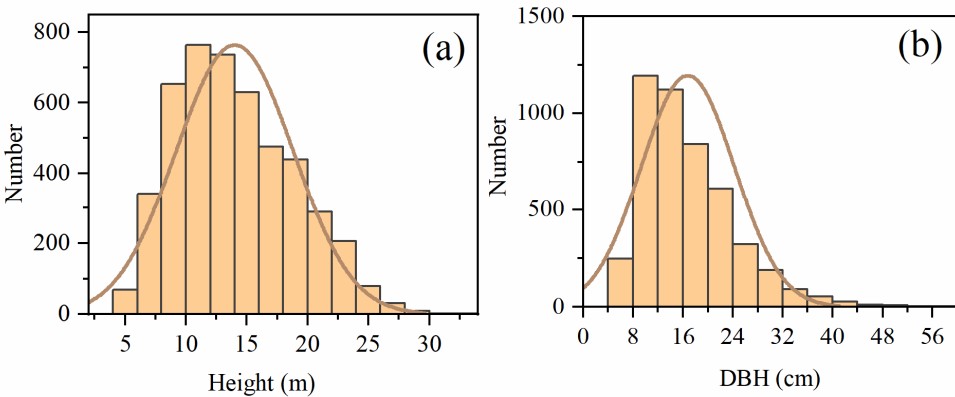

**Figure 2.** Histograms of individual tree (**a**) height and (**b**) DBH for the 62 sample plots.

**Table 1.** Statistical information for the 62 sample plots.

|  | Max | Min | Mean | Std |
|---|---|---|---|---|
| Stand mean height | 17.8 | 12.6 | 14.5 | 3.2 |
| Stand mean DBH | 16.8 | 11.4 | 13.4 | 2.9 |
| Slope(°) | 21.0 | 3.0 | 12.3 | 4.1 |

### 2.2.2. UAV-LiDAR Data and Preprocessing

The UAV-LiDAR data was acquired using a Feima D200 UAV equipped with a LiDAR200 system in August 2019–2022. Flights were conducted at an altitude of 80 m, with a flying speed of 5.0 m/s and a route overlap rate of 50%. The LiDAR sensor uses the

RIEGL mini VUX-1UAV to record multiple echoes (up to 5 times), with a wavelength of 905 nm and a laser divergence angle of $1.6 \times 0.5$ mrad. The UAV-LiDAR data was collected with a scanning angle of $\pm 45°$, an average point cloud density of 200 pts/m$^2$, and a point cloud accuracy of 15 mm. Three preprocessing steps were applied to the raw point cloud: (1) removal of isolated noise points; (2) classification between ground and non-ground points by an improved progressive triangular irregular network with a maximum terrain angle of 88°, an iteration angle of 8°, and an iteration distance of 1.4 m [34]; and (3) normalization of point cloud height by converting the point cloud elevation values to vegetation height. The data pre-processing was conducted using LiDAR360 V5.2 software (http://www.lidar360.com; accessed on 1 September 2023).

*2.3. Methods*

2.3.1. Overview of Methodology

The methodology used in this study includes (1) data preprocessing; (2) feature extraction of canopy height distribution (CHD) and traditional LiDAR-derived metrics; (3) AGB modeling using four groups of CHD features (original CHD features, original LiDAR metrics, fitted CHD features, and selected-LiDAR metrics) and six models (i.e., ML group: SVR, RF, Xgboost; DL group: 1D-CNN4, 1D-VGG16, and 1D-Resnet34); and (4) two-way analysis of variance (ANOVA) to explore the main effects and interactions between features and prediction models (Figure 3).

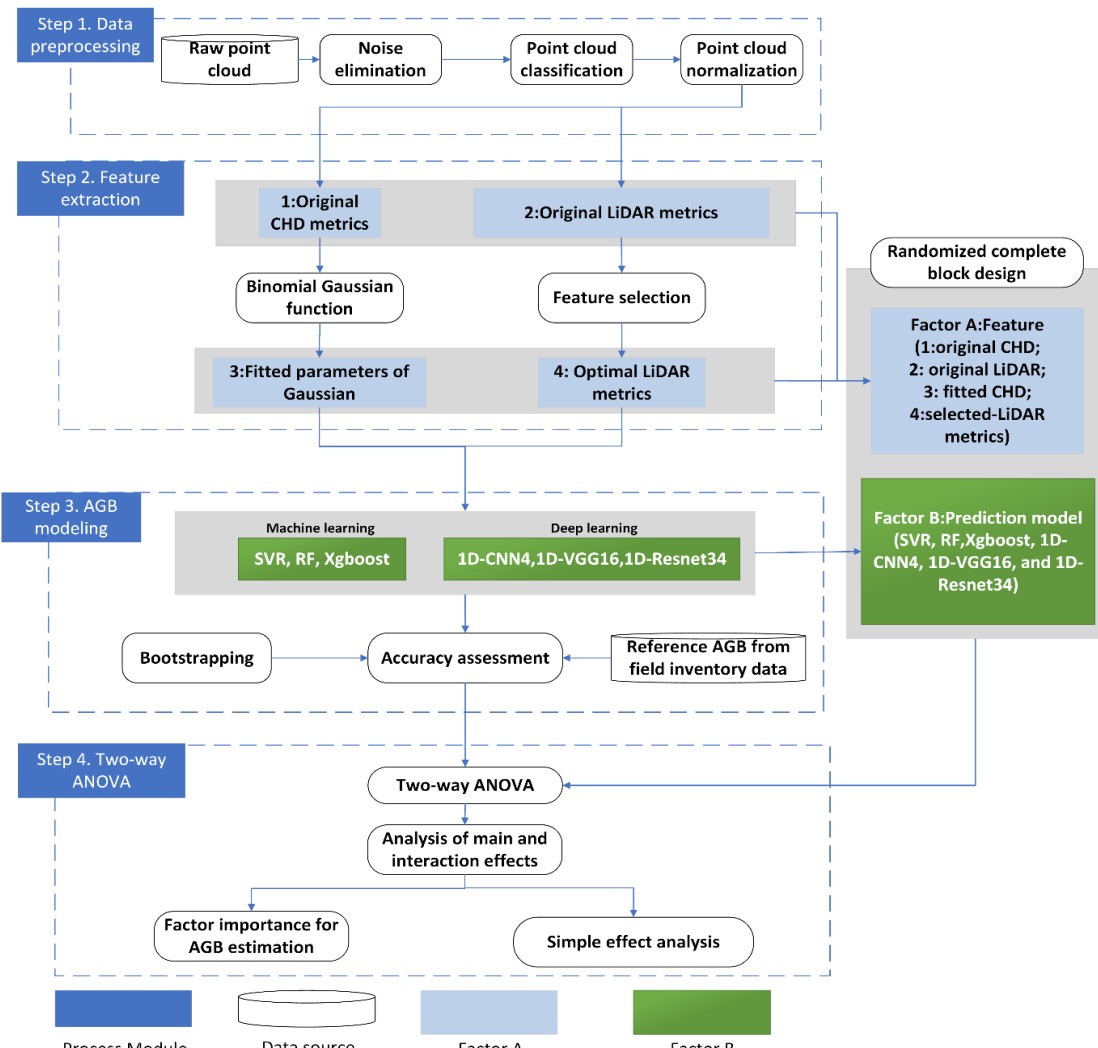

**Figure 3.** An overview of the workflow of this study.

### 2.3.2. Experiment Design

A completely randomized design (CRD) [35] was employed to analyze the main effects and interactions of two factors (i.e., features and prediction models) for the AGB estimation. The factorial treatments were designed as follows: The first factor was the feature with four levels: the original CHD features, the original LiDAR metrics, the parameters of fitted CHD curves (fitted CHD features), and selected LiDAR metrics. The second factor was the prediction model, including six models that can be classified into two model groups: machine learning models (i.e., SVR, RF, and Xgboost) and deep learning algorithms (i.e., 1D-CNN4, 1D-VGG16, and 1D-Resnet34). Therefore, 24 treatment (4 × 6) combinations were generated, and five replications were used for each treatment combination, resulting in 120 records for the two-way ANOVA.

### 2.3.3. Feature Extraction

- The extraction and curve fitting of CHD

In this study, CHD was represented by slicing all point clouds inside the plot at a height interval of 0.4 m and calculating the ratio of the number of points within each slice to the total number of points. The formula for calculating CHD is as follows:

$$f(h) = \int_0^{H_{max}} \frac{n_h}{N} dh \tag{2}$$

where $f(h)$ is a discrete CHD function and $h$ is the height; $n_h$ is the number of LiDAR echoes in a height interval; $N$ is the total number of point clouds in the sample plot; and $h$ is the sampling height interval.

The Gaussian function was used to model the CHD extracted with Equation (3) for each sample plot. The Gaussian function is expected to be able to describe stands with significant canopy stratifications due to their multiple peaks. This function consists of three parameters, denoting the maximum return ratio ($a$) at the canopy height of the curve peak ($b$) and the standard deviation ($c$). These parameters contain a large amount of information about the canopy, which can well characterize the structure of the canopy. Due to the strong association between the canopy's structure and AGB, these parameters could be beneficial for estimating AGB. The Gaussian function is expressed as follows:

$$f(h) = \sum_{i=1}^{n} a_i e^{\frac{-(h-b_i)^2}{c_i^2}} \tag{3}$$

where $h$ is the canopy height; $n$ is the number of peaks to fit, which was set to $n = 2$ in this study (i.e., a bimodal Gaussian function) because the natural secondary forests in northeastern China are either single-story stands or two-story stands; $a_i$ is the maximum point cloud ratio at the height of the curve peak; $b_i$ is the center coordinate of the peak, representing the canopy height where the slice with the most echoes is located; and $c_i$ is the standard deviation representing the width of the curve. The parameters of the fitted CHD curves were used as fitted CHD features in the AGB estimation model. Those parameters need to be estimated from the LiDAR data for each plot.

- The extraction and selection of LiDAR metrics

To avoid ground and low shrub points, the points below 2 m were not involved in the calculation. 101 LiDAR metrics [16] were extracted from the normalized point cloud, including 46 height metrics, 42 intensity metrics, 10 density metrics, and three forest metrics. The details of all LiDAR-derived metrics are listed in Table A2. To determine the optimal feature number, the Recursive feature elimination (RFE) algorithm was used to evaluate the rank of features. RFE is an approach to wrapper feature selection where the optimal subset of features is obtained by iteratively removing the less important features. We embedded different numbers of LiDAR metrics into the six models used in this study to find the best subset of features.

2.3.4. AGB Modeling

The forest AGB modeling in this study consisted of the following three steps:

(I) Three ML models (i.e., SVR, RF, and Xgboost) and three DL models (i.e., 1D-CNN4, 1D-VGG16, and 1D-Resnet34) were utilized to establish the AGB models. The SVR uses a nonlinear kernel function to determine an ideal regression plane by minimizing the distance from the data to that plane [36,37]. The RF combines several weak decision tree classifiers to provide the model with outstanding accuracy and generalization [38,39]. Xgboost is an improved gradient boosting algorithm. It is one of the boosting ensemble learning algorithms that improves prediction accuracy by combining multiple weak learners (i.e., decision trees).

CNN, one of the representative deep learning algorithms, utilizes several convolutional layers to extract useful features [40]. In the 1D-CNN4 model, the Relu activation function enables the model to handle nonlinear tasks; a maximum pooling with three windows was adopted for downsampling to avoid over-fitting; the flattened layer was used to flatten the multidimensional data into one dimension, acting as a transition between the convolutional layer and the fully connected layer; and the dense layer (i.e., the fully connected layer) was used for data output. The 1D-VGG16 algorithm is a very excellent model in the field of classification, consisting of 13 convolutional layers and three fully connected layers [30]. Resnet is composed of many residual blocks, which can handle hyper-deep (>1000) convolutional networks through the residual blocks and well solve the gradient disappearance/gradient explosion during model fitting [32]. Resnet introduces a batch normalization layer instead of the dropout layer used in VGG to speed up training and improve fitting accuracy. This study applied 34 convolutional layers of Resnet (i.e., Resnet34) through trial and error. Since VGG and Resnet were initially designed to classify 2D images, we redesigned them to accommodate a 1D dataset by changing 2D-Convolution to 1D-Convolution. The three DL models are shown in Figure 4.

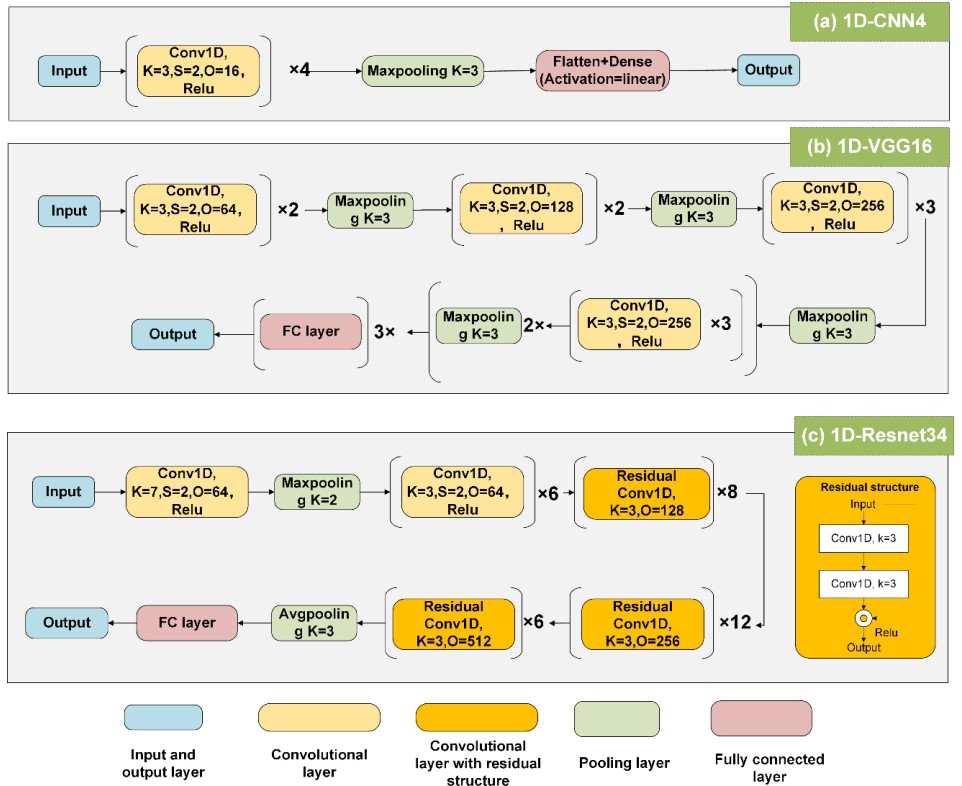

**Figure 4.** The three DL structures were designed in this study. (**a**) Custom 1D-CNN4 structure with four convolutional layers; (**b**) VGG16 network structure; (**c**) Resnet structure with 34 convolutional layers. Conv1D is one-dimensional convolution; K is kernel size; S is stride; and O is the output channel.

The hyperparameter tuning of ML models, especially DL models, is complex. We adopted a Bayesian-based approach [41] for hyperparameter optimization. The method uses a Gaussian process as the prior of the optimization function, computes the posterior distribution, and updates the Gaussian process, iterating the above process to maximize the optimization function. The hyperparameter tuning was performed in the Optuna package for Python (https://optuna.org/; accessed on 1 September 2023). It finds optimal hyperparameters and calculates their importance. Python 3.7, Sklearn, and TensorFlow 2.2 were used for modeling in this study. The operating system is Windows 10, the CPU is an R7 3700X, the graphics card is a 1080ti, and the memory is 32GB. The detailed hyperparameters of the six prediction models are summarized in Table 2.

**Table 2.** The hyperparameters range for the six biomass prediction models.

| Model Group | Model | Parameters Range |
|---|---|---|
| Machine learning | SVR | Kernels = ['linear kernels', 'ploy', 'rbf'], C = [1, 10], Gamma = ['auto', 'scale'] |
| | RF | n_estimators = [100, 1000], max_features = [1, 10], max_depth = [1, 30] |
| | Xgboost | n_estimators = [100, 1000], gamma = [0.1, 1], max_depth = [1, 30], seed = [1, 20] |
| Deep learning | 1D-CNN4, 1D-VGG16, 1D-Resnet34 | Epoch = [10, 1000], Batch size = [10, 50], Learning Rate = [0.0001, 0.001], Optimizer = ['Adam', 'SGD', 'RMSProp'] |

Note: The Kernels, C, and Gamma are the kernel function type, the regularization factor, and the kernel function coefficient in the SVR model, respectively. The n_estimators, max_features, and max_depth are the number of trees, the number of tree features, and the maximum depth of the tree in the RF model, respectively. Seed is the number of random seed points in Xgboost. Epoch, Batch size, Learning Rate, and Optimizer are the number of iterations, number of data per batch, learning rate, and optimizer type in CNNs, respectively.

Because the sample size (number of plots) in this study was modest, the prediction accuracy of the six models was evaluated using the leave-one-out cross-validation (LOOCV) method [42], which is commonly used for AGB model evaluation. It also improves the generalization ability of the model and mitigates overfitting [43]. Model accuracy evaluation metrics were $R^2$, RMSE (Mg/ha), and rRMSE between reference and predicted plot AGB.

(II) Five replications of accuracy metrics (i.e., $R^2$, RMSE, and rRMSE) were prepared for each treatment combination of features and prediction models ($4 \times 6 = 24$ treatment combinations in total if taking one replication as an example). Bootstrapped samples were generated using the bootstrapping procedure to satisfy the ANOVA assumptions of randomness and replication [44]. The 62 samples were randomly selected with replacement, and LOOCV in Step (I) above was applied to evaluate the accuracy of a specific model using one feature set. A 100-times permutated bootstrap was conducted, and the average of the accuracy metrics (i.e., $R^2$, RMSE, and rRMSE) that 100 bootstrapped samples produced were applied as the accuracy to avoid the influence of extreme values [26]. The above procedure was repeated five times (a total of 500-times bootstrapping) to obtain five sets of $R^2$, RMSE, and rRMSE for the subsequent ANOVA for one feature set and one prediction model. The logic of the bootstrapping procedure is shown in Figure 5.

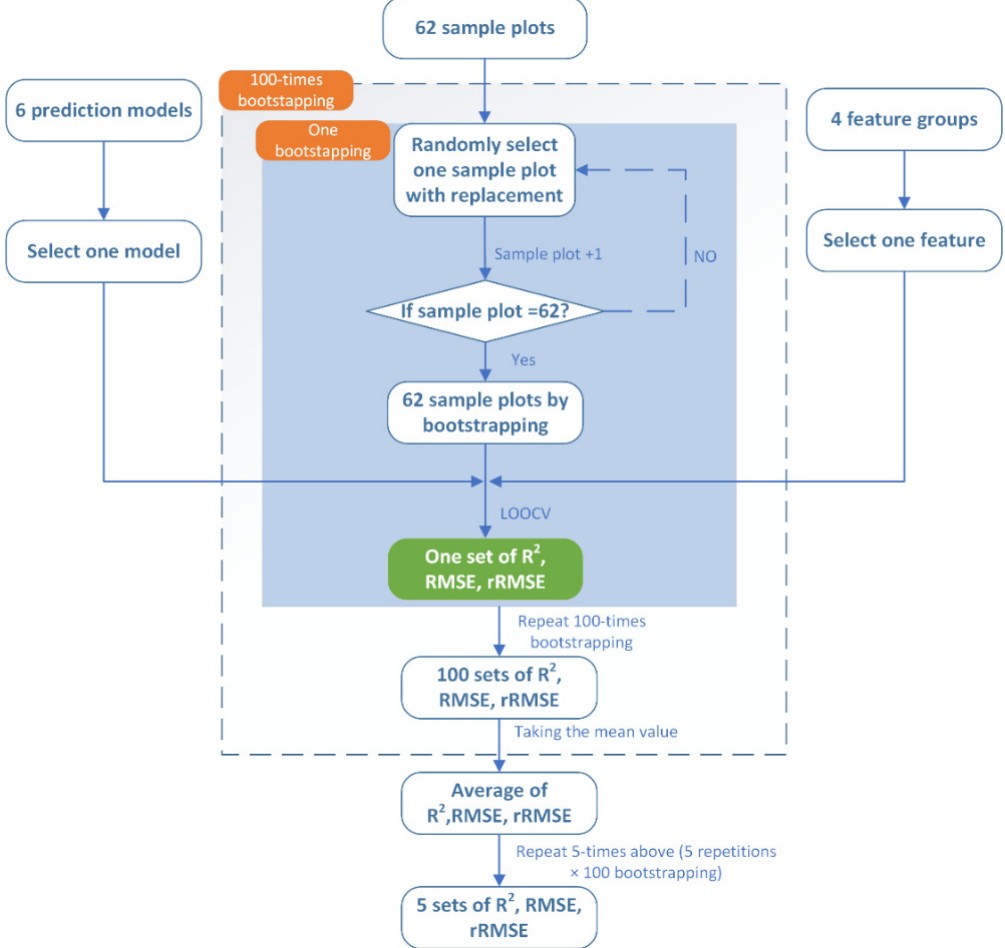

**Figure 5.** The flowchart of the bootstrapping procedures in this study, with one prediction model and one feature as examples.

(III) To quantify the impacts of factors and their interactions on the model accuracy evaluation metrics (i.e., $R^2$, RMSE, and rRMSE), a two-way ANOVA was conducted, and the two factors (i.e., feature and prediction model) were considered fixed effects. Eta-squared ($\eta^2$) [45], which is the ratio of the sum of squares (SS) of the factors to the total SS, was calculated to estimate the effect size of each main effect and interaction effect in the ANOVA model. A simple effects analysis [46] was performed to further explore the interaction between two factors. The design of the simple effects analysis is shown in Tables A3 and A4.

## 3. Results

### 3.1. Feature Extraction and Hyperparameter Tuning

#### 3.1.1. The Extraction and Curve Fitting of CHD

The Gaussian function (Equation (3)) was used to fit the CHD curves from the LiDAR data. Using two sample plots as examples, Figure 6 illustrates the unimodal CHD curve for a single-story stand (Figure 6a) and the bimodal CHD curve for a two-story stand (Figure 6b). For the single-story stand, the CHD curve reached its inflection point at the canopy height of 8 m, indicating a dramatic rise in the quantity of laser hits on the vegetation. The three parameters of the CHD curve were estimated as follows: The maximum return ratio was 0.046 ($a_1$), which corresponded to a canopy height of 10.080 m ($b_1$), and the curve width was 2.910 ($c_1$). This canopy height indicated where the greatest number of laser hits were obtained (Figure 6a).

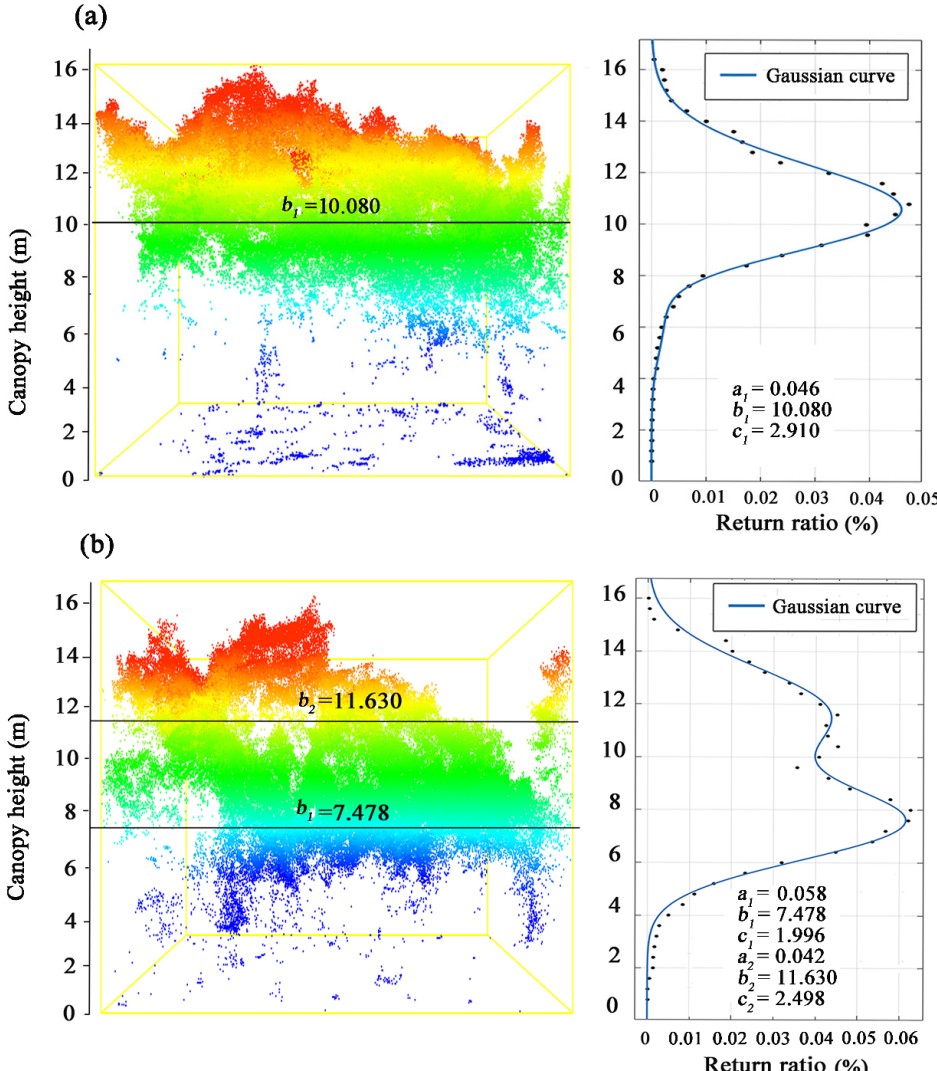

**Figure 6.** The CHD curves and parameters fitted by a bimodal Gaussian function for (**a**) a single-story sample plot, and (**b**) a two-story sample plot. Note: the different colors indicate the different canopy heights.

For the two-story stand, the CHD curve exhibited two peaks (Figure 6b). As the curve reflection occurred at the canopy height of about 4 m, the number of point cloud returns increased dramatically. The first return peak corresponded to the canopy height of 7.478 m ($b_1 = 7.478$), while the second peak occurred at a height of 11.630 m ($b_2 = 11.630$). The first peak was caused by the echo hits on the canopy of lower-story trees and the branches of taller trees, while the second peak was caused by the echo hits only on the canopy of taller trees. This was likely the reason why the return number of the first peak ($a_1 = 0.058$) was higher than that of the second peak ($a_2 = 0.042$).

### 3.1.2. Extraction and Selection of LiDAR Metrics

The 101 extracted LiDAR features (see Table A2) were ranked using the RFE method. Figure 7 displays the accuracy of AGB estimation for the three ML models and the three DL models with increasing feature numbers. It was found that when the number of LiDAR metrics increased from 1 to about 16, the accuracy of the six models improved dramatically. In general, all models achieved good performances with a feature number of 16. The model accuracies began to level off with a small fluctuation after the feature number of 16. Increasing the feature number had no significant effect on improving the model's accuracy but would increase the computational burden. Thus, 16 LiDAR features were selected to

reduce feature redundancy. The details and ranking (based on RFE) of the selected features are described in Table A5, respectively.

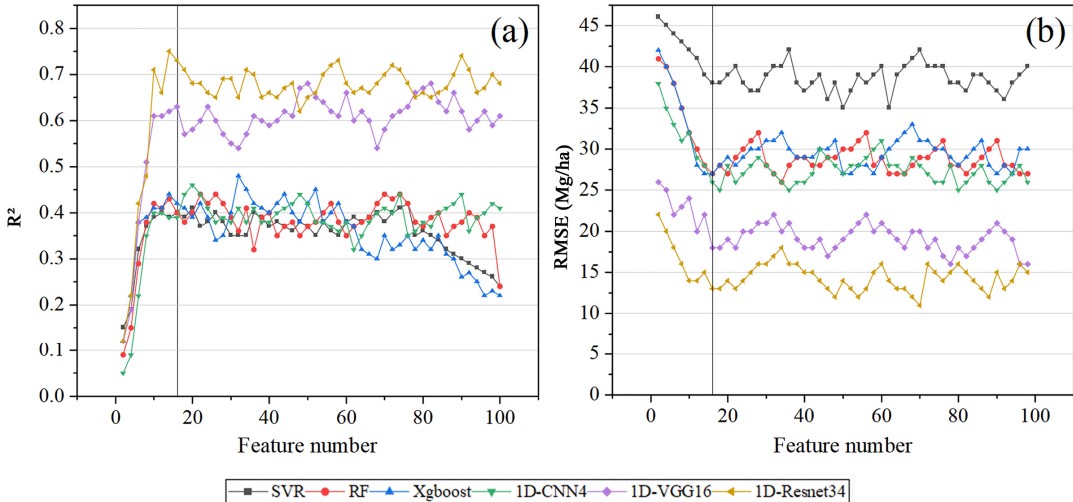

**Figure 7.** The accuracy of the six models with the change of feature number for LiDAR metrics: (**a**) $R^2$; (**b**) RMSE.

### 3.1.3. Hyperparameter Tuning

Table 3 shows the best combination of hyperparameters for the six models under four feature sets (i.e., original CHD, original LiDAR metrics, fitted CHD, and selected LiDAR metrics). For the SVR model, the hyperparameters for the four feature sets were held constant. Gamma = 'auto', C = 1, and kernel = 'rbf' were found to be the optimal combination. For the RF model, the hyperparameters for the four sets of features varied within a small range. For example, the n_estimators for the four groups were 852, 981, 736, and 1030, respectively. For the three DL models, the learning rate was set at $10^{-4}$. The Root Mean Square Prop (RMSProp) and stochastic gradient descent (SGD) optimizers yielded the best performance. The batch sizes were mostly 10–20, and the number of epochs was set between 600 and 1300.

**Table 3.** The determined hyperparameters of the six models using different feature sets.

| Model | Hypermeter | Original CHD | Fitted CHD | Original LiDAR | Selected-LiDAR |
|---|---|---|---|---|---|
| SVR | Gamma | auto | auto | auto | auto |
| | C | 1 | 1 | 1 | 1 |
| | Kernels | rbf' | rbf' | rbf' | rbf' |
| RF | Max_depth | 10 | 10 | 9 | 12 |
| | Max_feature | 8 | 7 | 8 | 9 |
| | n_estimators | 852 | 981 | 736 | 1030 |
| Xgboost | Gamma | 0.1 | 0.1 | 0.10 | 0.1 |
| | Seed | 2 | 3 | 5 | 4 |
| | Max_depth | 15 | 16 | 14 | 20 |
| | n_estimators | 1157 | 1036 | 826 | 972 |
| 1D-CNN4/ | | 0.0001/ | 0.0001/ | 0.0002/ | 0.0002/ |
| 1D-VGG16/ | Learning_rate | 0.0003/ | 0.0002/ | 0.0002/ | 0.0001/ |
| 1D-Resnet34 | | 0.0001 | 0.0001 | 0.0001 | 0.0001 |

Note: SGD: stochastic gradient descent; RMSProp: root mean square prop.

### 3.2. The Performance of AGB Prediction Models

The performances of the three ML models (SVR, RF, and Xgboost) and three DL models (1D-CNN4, 1D-VGG16, and 1D-Resnet34) were tested for AGB estimation using

selected/non-selected features (Table 4). Group I showed the AGB estimation performance of the three ML models based on the original CHD and LiDAR features. It was found that the original CHD and LiDAR features did not exhibit significant differences in the machine learning models. Group II showed the AGB estimation performance of the three ML models with CHD fitted by a bimodal Gaussian function and RFE-selected LiDAR metrics. The ML models based on the fitted CHD obviously performed better than those based on the selected LiDAR metrics. The AGB estimation accuracy reached the highest in the combination of the RF model and fitted CHD features ($R^2$ = 0.50, RMSE = 19.42 Mg/ha, rRMSE = 0.16). Compared with the original features, the fitted CHD and selected LiDAR metrics significantly improved the prediction accuracy and efficiency of the model (i.e., running time).

**Table 4.** The performances of the six AGB prediction models using original/fitted CHD features and original/selected LiDAR metrics.

| Group | Feature (Feature Number) | Model | $R^2$ | RMSE (Mg/ha) | rRMSE | Runtime |
|---|---|---|---|---|---|---|
| Group I | Original CHD (41–69) | SVR | 0.22 | 29.89 | 0.35 | 4 min 18 s |
| | | RF | 0.23 | 31.18 | 0.28 | 10 min 20 s |
| | | Xgboost | 0.20 | 30.54 | 0.28 | 10 min 33 s |
| | Original LiDAR metrics (101) | SVR | 0.24 | 38.37 | 0.34 | 6 min 12 s |
| | | RF | 0.24 | 27.15 | 0.27 | 12 min 47 s |
| | | Xgboost | 0.22 | 27.41 | 0.27 | 12 min 03 s |
| Group II | Fitted CHD (6) | SVR | 0.41 | 21.05 | 0.20 | 3 min 50 s |
| | | RF | 0.50 | 19.42 | 0.16 | 6 min 47 s |
| | | Xgboost | 0.48 | 20.33 | 0.18 | 6 min 38 s |
| | Selected LiDAR metrics (16) | SVR | 0.39 | 26.33 | 0.24 | 4 min 47 s |
| | | RF | 0.42 | 21.45 | 0.21 | 7 min 30 s |
| | | Xgboost | 0.41 | 22.36 | 0.22 | 7 min 41 s |
| Group III | Original CHD (41–69) | 1D-CNN4 | 0.21 | 33.21 | 0.29 | 26 min 35 s |
| | | 1D-VGG16 | 0.45 | 24.76 | 0.22 | 42 min 56 s |
| | | 1D-Resnet34 | 0.48 | 21.81 | 0.19 | 56 min 20 s |
| | Original LiDAR metrics (101) | 1D-CNN4 | 0.41 | 26.05 | 0.23 | 32 min 08 s |
| | | 1D-VGG16 | 0.61 | 18.85 | 0.17 | 45 min 18 s |
| | | 1D-Resnet34 | 0.68 | 13.14 | 0.12 | 62 min 26 s |
| Group IV | Fitted CHD (6) | 1D-CNN4 | 0.42 | 21.57 | 0.21 | 17 min 08 s |
| | | 1D-VGG16 | 0.65 | 18.55 | 0.17 | 36 min 47 s |
| | | 1D-Resnet34 | 0.80 | 9.58 | 0.09 | 48 min 11 s |
| | Selected LiDAR metrics (16) | 1D-CNN4 | 0.40 | 23.08 | 0.22 | 19 min 06 s |
| | | 1D-VGG16 | 0.61 | 19.06 | 0.18 | 46 min 27 s |
| | | 1D-Resnet34 | 0.71 | 12.72 | 0.11 | 52 min 33 s |

Note: Model accuracy is the mean value of 100 bootstraps. Machine learning group: SVR, RF, and Xgboost. Deep learning group: 1D-CNN4, 1D-VGG16, and 1D-Resnet34. The feature number of original CHD varied based on the canopy height of the sample plots.

Group III illustrated the performance of DL models for AGB estimation using the original CHD and LiDAR features. It showed the opposite results compared with the ML models: the DL models with the original LiDAR feature obtained significantly higher accuracies than those with the original CHD features. This is because the DL benefited from a large number of LiDAR features under the same conditions. Thus, the 101 original LiDAR features worked better than the 41–69 original CHD features for the DL models. Group IV showed the performance of the DL models using the fitted CHD and the selected LiDAR features. The fitted CHD features significantly outperformed the selected LiDAR metrics in 1D-VGG16 and 1D-Resnet34, especially in the 1D-Resnet34 model, of which the accuracy reached the highest ($R^2$ = 0.80, RMSE = 9.58 Mg/ha, rRMSE = 0.09). In addition,

the accuracies of the DL models with the original and feature-selected LiDAR features were very similar. In contrast, the accuracies of the DL models with the original and fitted CHD features were quite different. It indicated the obvious efficiency of fitting CHD features compared to the original CHD features. In general, the DL models are less efficient than the ML models.

### 3.3. Two-Way ANOVA

To investigate the main effects of the two factors (i.e., feature and prediction model) and their interactions on the prediction accuracy ($R^2$, RMSE, and rRMSE) of the AGB estimation, a two-way ANOVA was performed (Table 5). The main effects of the feature and prediction models and their two-way interactions were statistically significant (*p*-value < 0.05). The highest SS (a substantial contribution to the explained variance) was reported for the prediction model, explaining 63.8%, 51.5%, and 56.3% of the variation in $R^2$, RMSE, and rRMSE ($\eta^2$ in Table 5), respectively. The feature also had large effects on the variation of $R^2$ ($\eta^2$ of 30.6%), RMSE ($\eta^2$ of 37.2%), and rRMSE ($\eta^2$ of 35.4%). On the other hand, the interaction between the feature and predictive model had minimal effects ($\eta^2$ for $R^2$, RMSE, and rRMSE are 5.6%, 11.3%, and 8.3%, respectively).

**Table 5.** Two-way ANOVA for the main effects and interactions of the two factors.

| Factor | df | $R^2$ | | | RMSE | | | rRMSE | | |
|---|---|---|---|---|---|---|---|---|---|---|
| | | SS | $\eta^2$ (%) | *p*-Value | SS | $\eta^2$ (%) | *p*-Value | SS | $\eta^2$ (%) | *p*-Value |
| Feature | 3 | 1.04 | 30.6 | <0.0001 | 1832.09 | 37.2 | <0.0001 | 0.17 | 35.4 | <0.0001 |
| Model | 5 | 2.17 | 63.8 | <0.0001 | 2535.64 | 51.5 | <0.0001 | 0.27 | 56.3 | <0.0001 |
| Feature × Model | 15 | 0.19 | 5.6 | <0.0001 | 552.87 | 11.3 | <0.0001 | 0.04 | 8.3 | <0.0001 |

Since $R^2$ and RMSE presented the same trend in Table 5, we focused on RMSE and rRMSE in subsequent analysis. Figure 8 shows the interactions between features and prediction models on RMSE and rRMSE which are presented in two ways (i.e., Feature × Model, and Model × Feature). Figure 8 shows a clear interaction between the features and the prediction models. The RMSE and rRMSE of the fitted CHD feature were significantly lower than other features (e.g., original CHD, original LiDAR, and selected LiDAR), which indicated an effective feature set for the NSFs. The 1D-Resnet34 model exhibited the best performance, with a significantly lower RMSE and rRMSE than the other five models. The second-best model was the 1D-VGG16, which showed lower RMSE and rRMSE than the other four models (i.e., SVR, RF, Xgboost, and 1D-CNN4). On the other hand, RF, Xgboost, and 1D-CNN4 performed similarly. The SVR model performed worst among the six models.

Since the interaction between the feature and prediction model was significant, the simple effects of the treatment combinations should be reported instead of the main effects of the CHD feature and prediction model. Tukey's test was performed based on the mean of RMSE and rRMSE to test whether the models/features differed significantly under different features/models. From Figure 9, we found that Tukey's tests based on RMSE and rRMSE have the same trend for most cases. For each feature set, the differences among RF, Xgboost, and 1D-CNN4 models were not significant in most cases, while 1D-VGG16 and 1D-Resnet34 were significantly better than the four other models (Figure 9a,c). The fitted CHD significantly outperformed both the original CHD and the original LiDAR for different models; however, for some models (e.g., in 1D-VGG16 and 1D-Resnet34), the fitted CHD and the selected LiDAR were not significantly different (Figure 9b,d).

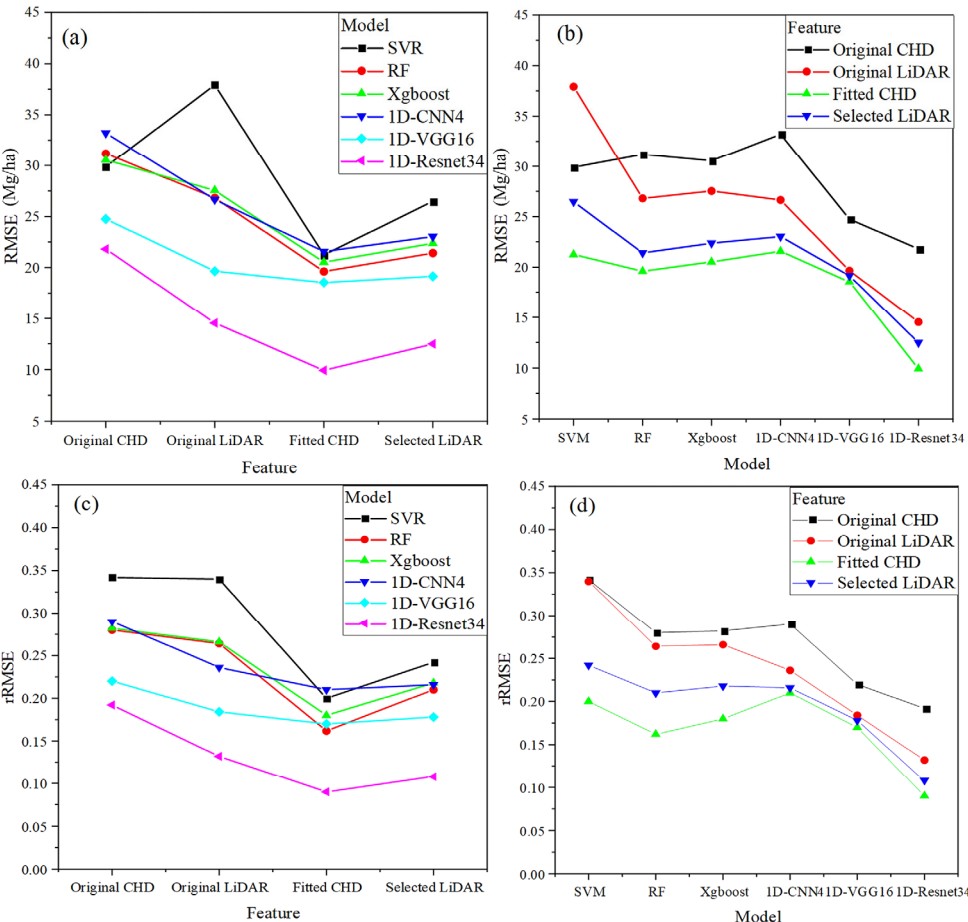

**Figure 8.** The interaction plots for RMSE and rRMSE: (**a**) RMSE: Feature × Model; (**b**) RMSE: Model × Feature; (**c**) rRMSE: Feature × Model; (**d**) rRMSE: Model × Feature.

To understand the contributions of the feature and prediction models, we organized 18 contrasts (details in Table A4). Table 6 showed that most interesting contrasts were statistically significant ($p$ values $< 0.05$), except for contrasts 17 and 18. It was found (from contrast ID:1–4) that the fitted CHD and the selected LiDAR significantly outperformed the original CHD and LiDAR metrics for both DL and ML models. For example, if the DL models were conducted, the RMSE and rRMSE would decrease by 9.91 Mg/ha and 7.73%, respectively, using the fitted CHD features compared to those using the original CHD features. The improvement of the fitted CHD for model accuracy was more significant compared with selected LiDAR metrics in the same model group (contrast ID: 1 vs. 2; ID: 3 vs. 4). The fitted CHD significantly outperformed the selected LiDAR metrics in both ML and DL models, and the improvement was more remarkable in the ML models than in the DL models (contrast ID: 5–6). For example, if the fitted CHD features were applied, the rRMSE would decrease by 4.27% for ML models and only 1% for DL models compared to those using traditionally selected LiDAR features. It also indicated that DL models could be a remedy for the AGB prediction accuracy if users still intend to use selected LiDAR features. From the model contrasts (contrast ID: 7–10), the DL models obviously performed better than the ML models regardless of the features; the improvement is larger when using original and selected LiDAR features than when using original and selected CHD features.

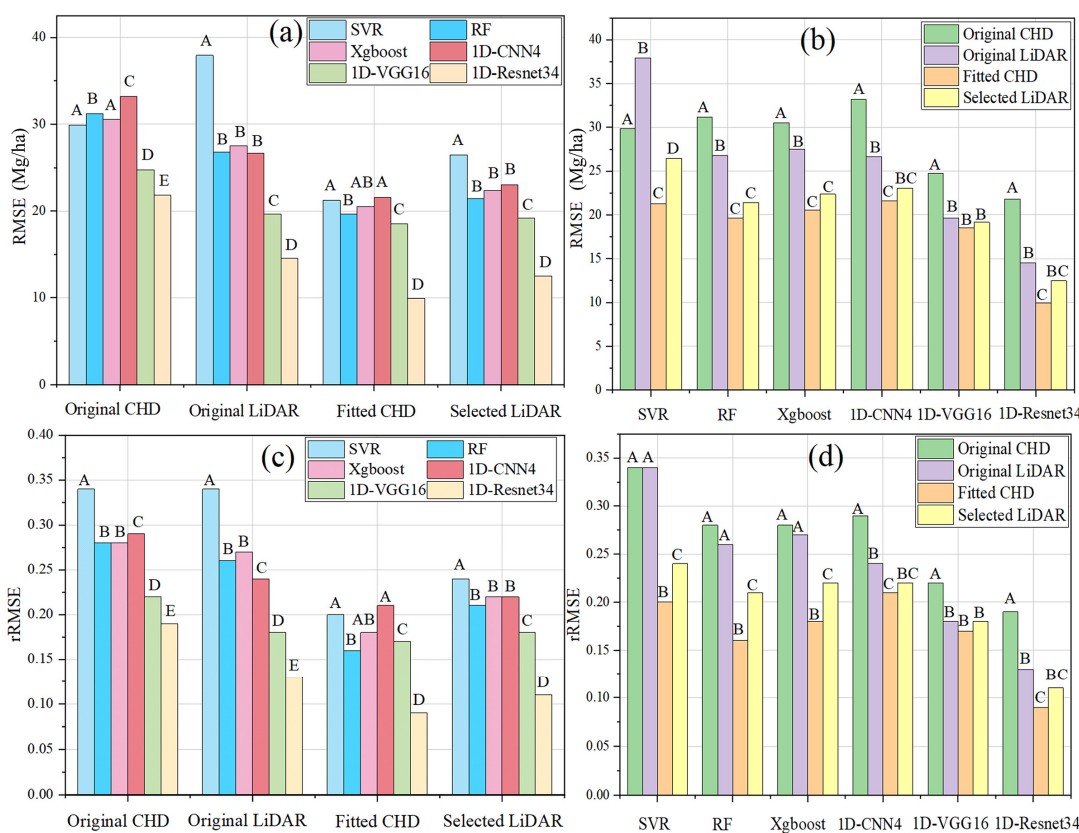

**Figure 9.** Tukey's test based on the mean of RMSE and rRMSE. (**a**) RMSE: comparison among six models for each feature; (**b**) RMSE: comparison among the four feature sets for each model; (**c**) rRMSE: comparison among six models for each feature; (**d**) rRMSE: comparison among the four feature sets for each model. Different capital letters indicate significant differences.

**Table 6.** Simple effects for RMSE (Mg/ha) and rRMSE (%) values.

| Contrast ID | Simple Effects | Estimate (RMSE/rRMSE) | *p* Value (RMSE/rRMSE) |
|:---:|:---:|:---:|:---:|
| 1 | Original vs. fitted CHD within DLs | 9.91/7.73 | <0.0001/<0.0001 |
| 2 | Original vs. selected LiDAR within DLs | 2.05/1.67 | <0.0001/0.0049 |
| 3 | Original vs. fitted CHD within MLs | 10.07/12.07 | <0.0001/<0.0001 |
| 4 | Original vs. selected LiDAR within MLs | 7.33/6.67 | <0.0001/<0.0001 |
| 5 | Fitted CHD vs. selected LiDAR within DLs | −1.54/−1.07 | 0.0001/0.0684 |
| 6 | Fitted CHD vs. selected LiDAR within MLs | −2.96/−4.27 | <0.0001/<0.0001 |
| 7 | DLs vs. MLs within Original CHD | −3.94/−6.73 | <0.0001/<0.0001 |
| 8 | DLs vs. MLs within Original LiDAR | −10.48/−10.60 | <0.0001/<0.0001 |
| 9 | DLs vs. MLs within Fitted CHD | −3.78/−2.40 | <0.0001/<0.0001 |
| 10 | DLs vs. MLs within Selected LiDAR | −5.20/−5.60 | <0.0001/<0.0001 |
| 11 | 1D-Resnet34 vs. 1D-CNN4 within Fitted CHD | −11.64/−12.00 | <0.0001/<0.0001 |
| 12 | 1D-Resnet34 vs. 1D-CNN4 within Selected LiDAR | −10.55/−10.80 | <0.0001/<0.0001 |
| 13 | 1D-Resnet34 vs. 1D-VGG16 within Fitted CHD | −8.62/−8.00 | <0.0001/<0.0001 |
| 14 | 1D-Resnet34 vs. 1D-VGG16 within Selected LiDAR | −6.67/−7.00 | <0.0001/<0.0001 |
| 15 | RF vs. SVM within Fitted CHD | −1.63/−3.80 | 0.0176/0.0003 |
| 16 | RF vs. SVM within Selected LiDAR | −5.06/−3.20 | <0.0001/0.0019 |
| 17 | RF vs. Xgboost within Fitted CHD | −0.91/−1.80 | 0.1806/0.0757 |
| 18 | RF vs. Xgboost within Selected LiDAR | −0.95/−0.80 | 0.1606/0.4268 |

Since the fitted CHD and selected LiDAR metrics performed better than the other original feature sets, comparisons within the DL or ML models were considered when these

two feature sets were applied. When using the fitted CHD and the selected LiDAR metrics, the 1D-Resnet34 model significantly outperformed other DL models (contrast ID: 11–14). The improvement was more significant from 1D-CNN4 than from 1D-VGG16, no matter which feature set was used. For the ML models, the RF model significantly outperformed the SVM model but did not show apparent advantages over the Xgboost model (contrast ID: 15–18).

## 4. Discussion

### 4.1. CHD vs. LiDAR Metrics

Our study confirmed the reliability of AGB estimation based on UAV-LiDAR for natural secondary forests with high canopy closure. The LiDAR metrics and CHD features were extracted. The LiDAR metrics were selected based on their importance for estimating AGB, similar to the metrics identified in other studies [47–49]. Among them, the percentiles of height and the height mean were the most selected features. For example, Wang et al. extracted the height percentile, height mean, canopy cover, and other features from the discrete LiDAR with an $R^2$ of 0.702 and an RMSE of 14.711 Mg/ha [47]. In addition to those common LiDAR metrics, we identified the height maximum as one of the most critical metrics. Furthermore, the LiDAR metrics selected for the six models were inconsistent. Even when utilizing the identical feature selection algorithm, it is not possible to identify a single LiDAR feature that can be considered to be best for all forest conditions and geographies [15,16,18,20].

Overall, the fitted CHD outperformed the feature-selected LiDAR metrics in the six models (Group II and Group IV in Table 4). There are two possible reasons: (1) Compared to the LiDAR metrics, the CHD features had fewer losses of canopy information and provided a more complete representation of the vertical structure of the canopy than other feature-selected LiDAR metrics [19]; and (2) since fitting the CHD curve with a bimodal Gaussian function accurately described the vertical structure of the canopy, this may have a strong correlation with AGB. The fitted parameter *a* in the CHD curve is the maximum return ratio (Figure 6), which represents the maximum density of the canopy in the vertical direction; parameter *b* is the canopy height corresponding to the maximum return ratio, which can represent the two-story canopy structure; and parameter *c* is the curve width, which represents the canopy width in the vertical direction. These three parameters comprehensively depict the three-dimensional structure of the canopy, which has a close relationship with AGB [20], and thus produces good performance in AGB estimation. Although Gaussian functions have been applied in the decomposition of full-waveform LiDAR data [50–52], they have rarely been applied to fit the canopy height features for the discrete LiDAR data. The CHD fitted by the binomial Gaussian function in this study was more reliable than those in previous studies. For example, Zhang et al. used Weibull-fitted CHD for AGB estimation and obtained an $R^2$ of 0.54, RMSE of 19.84 Mg/ha, and rRMSE of 23.25% [12]. We have demonstrated the efficacy of using the DL models (especially 1D-Resnet34) in conjunction with the fitted CHD features, resulting in excellent predictive outcomes. Our results indicated that the CHD features remained consistent in the AGB models and did not require a feature selection process [15,16,20], providing insight for describing the vertical structure of forests with a complex stand structure. Thus, it is expected to apply the CHD features fitted by a bimodal Gaussian function as a practical feature set for the AGB estimation of NSFs using UAV-LiDAR data in the future.

The sampling height interval is an essential parameter for characterizing the CHD features. We extracted CHD based on different sampling height intervals and tested the effect of the extracted CHDs at these height intervals (0.1 m–1.0 m) on the performance of the AGB model. The 0.4 m sampling interval identified was appropriate in this study (see Figure A1), similar to previous studies' findings [53,54].

Although the CHD was generated from the normalized LiDAR point cloud, the terrain may still affect the CHD trend due to the uncertainty of the normalization algorithm. Therefore, we investigated the effect of slope on the fitting of CHD curves. Figure 10a,b

demonstrate that the accuracy of CHD curve fitting decreases slightly when the slope is greater than 15°. Figure 10c,d illustrate the CHD curves of sample plots with the minimum and maximum slopes (i.e., 4° and 21°). It demonstrates that compared to the CHD curve with big slopes, the one with modest slopes was smoother and simpler. Particularly when the canopy height was less than 10 m, the CHD discrete points in plots with steep slopes changed drastically, causing some disturbance to the curve fitting.

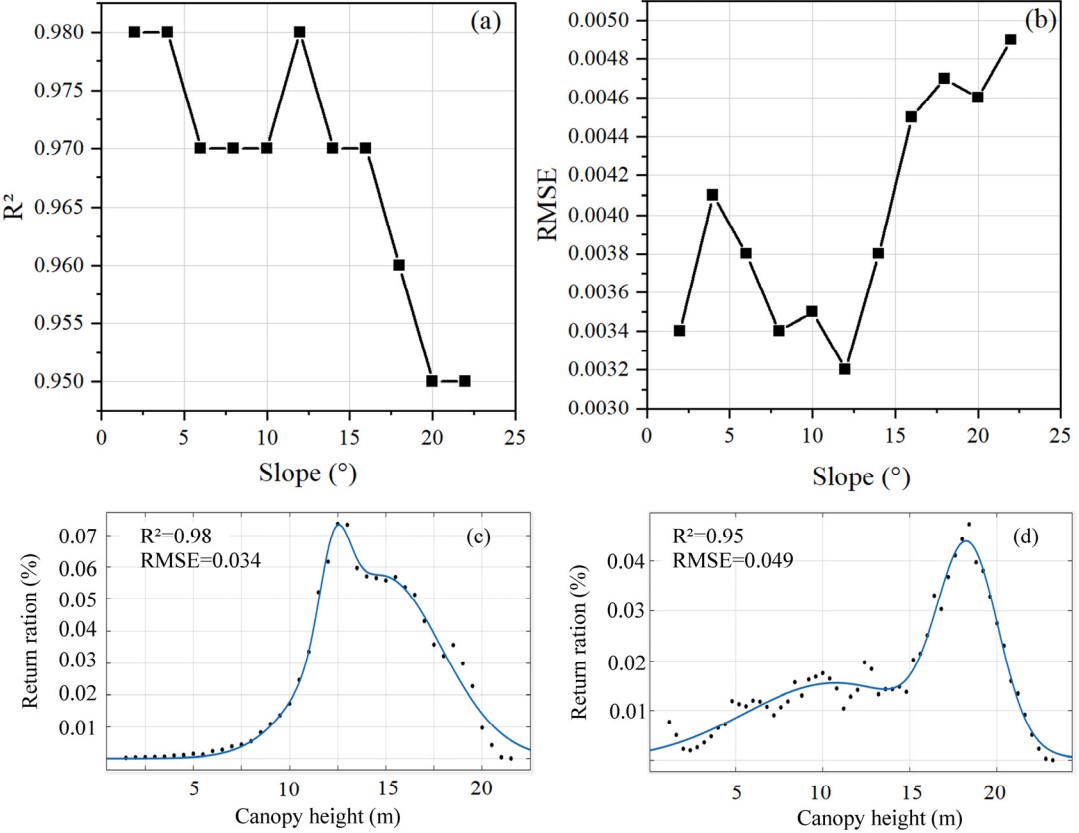

**Figure 10.** Effect of different slopes on CHD curve fitting. (**a**) $R^2$ of CHD curve fitting; (**b**) RMSE of CHD curve fitting; (**c**) CHD fitting curves for sample plots with the minimum slope of 4°; and (**d**) CHD fitting curves for sample plots with the maximum slope of 21°.

*4.2. Effect of Biomass Prediction Models*

The prediction model was an essential and critical factor for AGB estimation (Table 5), which is consistent with previous studies [55]. This may be due to the fact that a well-performing model has strong robustness and can be well supported for different data inputs. This study found that the differences between the two model groups were statistically significant, and the DL models significantly outperformed the ML models (Figure 9). It was found that the differences between the two model groups were statistically significant, and the DL models significantly outperformed the ML models. The DL model has been one of the most popular models in image recognition, image segmentation, and image classification, yet there is less research on forest attribute estimation [55]. In other studies, DL represented by CNNs has also shown excellent performance over the ML models [56–58]. In this study, the DL models consistently outperformed the ML models, regardless of the features used. This is mainly attributed to the powerful learning and weight-sharing capabilities of CNNs, which enable them to compute multiple features based on input data through convolutional layers. The shallow convolutional layers are mainly responsible for extracting texture and edge information, while the deeper convolutional layers can extract more abstract information that is useful for prediction [59]. This is one of the reasons that the Resnet with 34 convolutional layers outperformed the VGG with 16 layers in this

study. Although increasing the number of network layers is expected to improve the performance of CNN models, actual results show that model accuracy decreases in ultra-deep network layers due to the problem of gradient disappearance or explosion during the back-propagation process [32]. The unique residual structure of Resnet can solve the problem of gradient disappearance/explosion in ultra-deep networks, leading to greater accuracy in prediction than VGG16. To avoid overfitting, leave-one-out cross-validation was used for model validation, and a dropout layer was added to increase the generalization of the model.

Hyperparameters are critical for ML and DL models and must be finely tuned to achieve optimal performance. In this study, we utilized a Bayesian-based hyperparameter optimization method, which is more efficient than the traditional grid search approach [60]. Figure 11 shows the importance of the hyperparameters for all the models based on a Bayesian-based approach [41]. We found that batch size and epoch were the two most crucial hyperparameters in the DL model, surpassing the importance of optimizer type and learning rate. Batch size refers to the number of samples used for each training iteration, and an appropriate batch size can ensure gradient descent moves in the correct direction and yields desirable convergence results. However, if the batch size is too small, it can make convergence difficult and lead to underfitting [61]. Typically, a larger batch size produces better prediction results but increases the computational load. Therefore, finding a balance between computational efficiency and prediction accuracy is crucial. Meanwhile, epoch refers to the number of iterations, meaning that all training samples are passed forward and backward once. As the epoch increases, the model weights are continuously updated, and the model loss gradually decreases, leading to improved accuracy and stabilization. Therefore, a large epoch value (e.g., >200) is required according to the sample size. Additionally, the number of model trees (n_estimators) was vital for RF and Xgboost. As the number of trees increased, the stability of the model improved, but at the expense of an increased computational load. For SVR, the kernel function is the most important, as it determines the best regression hyperplane and, therefore, plays a critical role in prediction tasks.

### 4.3. Limitations and Future Research

Besides features and models, the AGB model accuracy using an area-based approach is significantly influenced by factors such as sample size, plot size, and plot shape [26,62]. When it comes to sampling, larger sample sizes tend to have smaller sampling errors and are therefore better representatives of the population than small sample sizes [63]. Large plot sizes and circular plot configurations have been found to be more effective for AGB estimation due to their ability to reduce edge effects [62]. In this study, we focused primarily on the impact of features and models on AGB estimation from a remote sensing perspective and did not delve into the influence of the sample data (e.g., sample size, plot size/shape). To test the validity of using canopy height diversity (CHD) for AGB estimation, we selected a complex mixed forest type (i.e., NSFs). To further demonstrate the wide applicability of CHD, we tested its performance in larch plantations (Table A6). The CHD performed better in the larch plantation compared to the NSFs (Table A6 vs. Table 4). This demonstrates the effectiveness of CHD in plantation forests. Diverse stand types could be examined for the generalizability of CHD in the future. In addition, high point cloud density is important to capture the CHD curve completely. The LiDAR sensors carried by the UAV can collect high point cloud density since the UAV has a lower flight altitude (e.g., 80 m in this study) and a lower flight speed (e.g., 5 m/s in this study) compared to ALS data. However, acquiring high-density point cloud data requires high flight costs. The effect of changing point cloud density on CHD curve fitting can be further explored in the future.

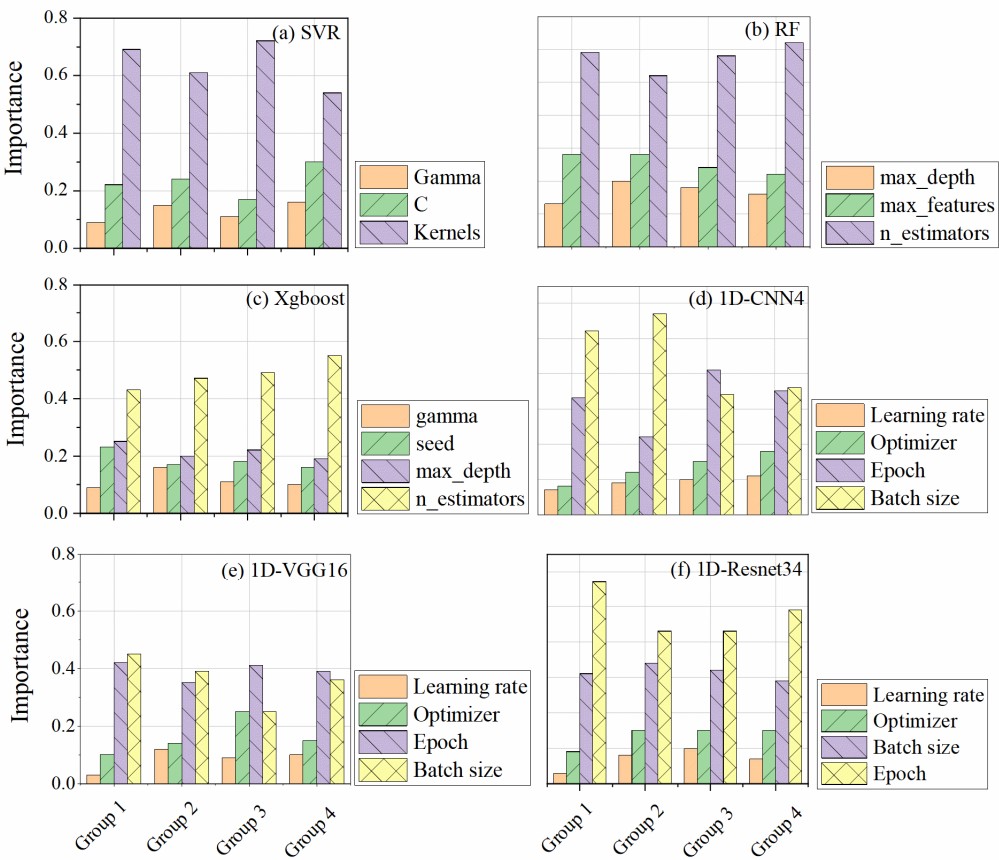

**Figure 11.** The importance and best combination of hyperparameters for the 3 ML models and the 3 DL models based on the Bayesian approach. Group1: original CHD; Group2: original LiDAR metrics; Group3: fitted CHD; Group4: selected-LiDAR metrics.

While 1D-CNNs offer superior classification accuracy, their utilization incurs specific considerations such as the need for large training datasets, significant time commitments, and extensive hyperparameter tuning. Conversely, traditional machine learning algorithms offer faster processing times. Hence, the prediction model should be carefully selected after balancing the prediction accuracy and computation load. Moreover, since the sample size of ground surveys in forestry is small, deep learning methods such as transfer learning [64], generative adversarial networks (GAN) [65], and meta-learning [66] that are suitable for small sample sizes should be fully explored in the future. To increase the interpretability of deep learning models, the SHapley Additive exPlanations (SHAP) method can also be considered to measure the feature contribution in the future [67].

## 5. Conclusions

This study examined the factors that affect area-based AGB estimation of NSFs using UAV-LiDAR data, including features and prediction models. It revealed that the prediction model had a greater impact on forest AGB estimation than the feature and the interaction between the model and the feature. No matter which features were applied, the DL models (i.e., 1D-CNN, 1D-VGG16, and 1D-Resnet34) outperformed the ML models (i.e., SVR, RF, and Xgboost), and the 1D-Resnet34 model achieved the highest accuracy using the fitted CHD feature ($R^2 = 0.80$, RMSE = 9.58 Mg/ha, rRMSE = 0.09).

Additionally, the parameters fitted to the CHD curves showed superior performance compared to traditional LiDAR metrics, which indicated that the bimodal Gaussian function was a highly effective method for fitting the CHD curves, providing accurate representations of the vertical distribution of canopy not only for single-storied stands but also for two-storied stands. The advantages of using the fitted CHD features over traditional

LiDAR metrics may allow for a simplification of the feature extraction and selection process for AGB estimation. The bimodal Gaussian function could also be applied to more complex stand structures, such as multi-layer mixed species stands, and needs to be validated in the future. This study would provide sound foundations for effective and accurate estimation of AGB in NSFs using appropriate features and models.

**Author Contributions:** Conceptualization, Z.Z. and Y.Z.; methodology, Y.M.; software, Y.M.; validation, Y.M., L.Z. and Z.Z.; formal analysis, Y.M.; investigation, Y.M.; resources, Z.Z. and Y.Z.; data curation, Y.M.; writing—original draft preparation, Y.M.; writing—review and editing, L.Z., J.I., Z.Z. and Y.Z.; visualization, Y.M.; supervision, L.Z. and J.I.; project administration, Y.M.; funding acquisition, Y.Z. and Z.Z. All authors have read and agreed to the published version of the manuscript.

**Funding:** This research was funded by the National Natural Science Foundation of China, "Multi-scale forest aboveground biomass estimation and its spatial uncertainty analysis based on individual tree detection techniques" (32071677), the Science and Technology Basic Resources Investigation Program of China (No. 2019FY101602-1), the National Forestry and Grassland Data Center-Heilongjiang platform (2005DKA32200-OH), and the National Natural Science Foundation of China, grant number (31870530). J. Im was partially supported by Korea Environment Industry & Technology Institute (KEITI) through Project for developing an observation-based GHG emissions geospatial information map, funded by Korea Ministry of Environment (MOE) (RS-2023-00232066).

**Data Availability Statement:** The data that support the findings of this study are available on request from the corresponding author, (Z.Z.). The data are not publicly available because it contains information that could compromise the privacy of research participants.

**Conflicts of Interest:** The authors declare no conflict of interest.

## Appendix A

**Table A1.** The parameters of additive biomass equations for different tree species and components used in this study [1,33].

| Tree Species | Component | $a_i$ | $b_i$ | Tree Species | Component | $a_i$ | $b_i$ |
|---|---|---|---|---|---|---|---|
| Korean pine | branch | −3.3911 | 2.0066 | Changbai larch | branch | −4.9082 | 2.5139 |
| | foliage | −2.6995 | 1.5583 | | foliage | −4.2379 | 1.8784 |
| | stem | −2.2319 | 2.2358 | | stem | −2.5856 | 2.4856 |
| white birch | branch | −5.7625 | 3.0656 | elm | branch | −3.0159 | 2.0328 |
| | foliage | −5.9711 | 2.5871 | | foliage | −3.4241 | 1.7038 |
| | stem | −2.8496 | 2.5406 | | stem | −2.2812 | 2.3766 |
| Manchurian ash | branch | −5.5012 | 2.9299 | Manchurian walnut | branch | −4.0735 | 2.4477 |
| | foliage | −5.2438 | 2.345 | | foliage | −5.0456 | 2.2577 |
| | stem | −3.4542 | 2.7104 | | stem | −2.6707 | 2.4413 |
| Mongolian oak | branch | −2.5856 | 2.4856 | Mono maple | branch | −2.2812 | 2.3766 |
| | foliage | −6.997 | 3.5220 | | foliage | −3.3225 | 2.2742 |
| | stem | −5.146 | 2.3185 | | stem | −3.3137 | 1.7074 |

Note: There are other tree species (e.g., mountain poplar and willow), which are so few in number that they are not included in the table.

**Table A2.** The description of LiDAR-derived metrics.

| Group | LiDAR Metrics | Description |
| --- | --- | --- |
| Height metrics (46) | elev_AAD | Average absolute deviation of point cloud height. |
| | elev_CRR | Canopy relief ratio of point cloud height. |
| | elev_GM_3rd | Generalized means for the third power. |
| | elev_cv | Coefficient of variation of point cloud height. |
| | elev_IQ | Elevation percentile interquartile distance. |
| | elev_kurt | Kurtosis of point cloud height. |
| | elev_Mmad | Median of median absolute deviation of point cloud height. |
| | elev_max | Maximum of point cloud height. |
| | elev_min | Minimum of point cloud height. |
| | elev_mean | Mean of point cloud height. |
| | elev_median | Median of point cloud height. |
| | elev_skew | Skewness of point cloud height. |
| | elev_std | Standard deviation of point cloud height. |
| | elev_var | Variance of point cloud height. |
| | elev_per_i | The height percentiles of point cloud with 5% height intervals. |
| | elev_AIH_i | Within a statistical cell, all normalized lidar point clouds are sorted according to the height and the cumulative heights of all points are calculated. The cumulative height of i % points is the statistical unit's Accumulated height percentile (AIH). |
| | elev_AIH_IQ | AIH interquartile distance. |
| Intensity metrics (42) | int_AAD | Average absolute deviation of point cloud intensity |
| | int _cv | Coefficient of variation of point cloud intensity |
| | int _IQ | Percentile interquartile distance of point cloud intensity |
| | int _kurt | Kurtosis of point cloud intensity |
| | int _Mmad | Median of median absolute deviation of point cloud intensity |
| | int _max | Maximum of point cloud intensity |
| | int _min | Minimum of point cloud intensity |
| | int _mean | Mean of point cloud intensity |
| | int _median | Median of point cloud intensity |
| | int _skew | Skewness of point cloud intensity |
| | int _std | Standard deviation of point cloud intensity |
| | int _var | Variance of point cloud intensity |
| | int_per_i | The intensity percentiles of point cloud with 5% intensity intervals |
| | int_AIH_i | Within a statistical cell, all point clouds are sorted according to the intensity and the cumulative intensity of all points are calculated. The cumulative intensity of i % points is the statistical unit's Accumulated height percentile (AIH) |
| Density metrics (10) | den_i | The proportion of returns in i th height interval, i = 1 to 10. |
| Forest metrics (3) | CC | Canopy cover |
| | GF | Gap fraction |
| | LAI | Leaf area index |

**Table A3.** The mean list for 24 treatments.

| | Deep Learning (DL) Models | | | Machine Learning (ML) Models | | |
|---|---|---|---|---|---|---|
| | **1D-CNN** | **1D-Resnet** | **1D-VGG16** | **RF** | **SVM** | **Xgboost** |
| Original CHD | $\mu1$ | $\mu2$ | $\mu3$ | $\mu4$ | $\mu5$ | $\mu6$ |
| Original LiDAR | $\mu7$ | $\mu8$ | $\mu9$ | $\mu10$ | $\mu11$ | $\mu12$ |
| Fitted CHD | $\mu13$ | $\mu14$ | $\mu15$ | $\mu16$ | $\mu17$ | $\mu18$ |
| Selected LiDAR | $\mu19$ | $\mu20$ | $\mu21$ | $\mu22$ | $\mu23$ | $\mu24$ |

**Table A4.** Design of contrasts of the simple effects.

| Contrast ID | Simple Effects | Contrast |
|---|---|---|
| 1 | Original vs. fitted CHD within DLs | $\frac{(\mu1+\mu2+\mu3)}{3} - \frac{(\mu13+\mu14+\mu15)}{3}$ |
| 2 | Original vs. selected LiDAR within DLs | $\frac{(\mu7+\mu8+\mu9)}{3} - \frac{(\mu19+\mu20+\mu21)}{3}$ |
| 3 | Original vs. fitted CHD within MLs | $\frac{(\mu4+\mu5+\mu6)}{3} - \frac{(\mu16+\mu17+\mu18)}{3}$ |
| 4 | Original vs. selected LiDAR within MLs | $\frac{(\mu10+\mu11+\mu12)}{3} - \frac{(\mu22+\mu23+\mu24)}{3}$ |
| 5 | Fitted CHD vs. selected LiDAR within DLs | $\frac{(\mu13+\mu14+\mu15)}{3} - \frac{(\mu19+\mu20+\mu21)}{3}$ |
| 6 | Fitted CHD vs. selected LiDAR within MLs | $\frac{(\mu16+\mu17+\mu18)}{3} - \frac{(\mu22+\mu23+\mu24)}{3}$ |
| 7 | DLs vs. MLs within Original CHD | $\frac{(\mu1+\mu2+\mu3)}{3} - \frac{(\mu4+\mu5+\mu6)}{3}$ |
| 8 | DLs vs. MLs within Original LiDAR | $\frac{(\mu7+\mu8+\mu9)}{3} - \frac{(\mu10+\mu11+\mu12)}{3}$ |
| 9 | DLs vs. MLs within Fitted CHD | $\frac{(\mu13+\mu14+\mu15)}{3} - \frac{(\mu16+\mu17+\mu18)}{3}$ |
| 10 | DLs vs. MLs within Selected LiDAR | $\frac{(\mu19+\mu20+\mu21)}{3} - \frac{(\mu22+\mu23+\mu24)}{3}$ |
| 11 | 1D-Resnet vs. 1D-CNN within Fitted CHD | $\mu14 - \mu13$ |
| 12 | 1D-Resnet vs. 1D-CNN within Selected LiDAR | $\mu20 - \mu19$ |
| 13 | 1D-Resnet vs. 1D-VGG16 within Fitted CHD | $\mu14 - \mu15$ |
| 14 | 1D-Resnet vs. 1D-VGG16 within Selected LiDAR | $\mu20 - \mu21$ |
| 15 | RF vs. SVM within Fitted CHD | $\mu16 - \mu17$ |
| 16 | RF vs. SVM within Selected LiDAR | $\mu22 - \mu23$ |
| 17 | RF vs. Xgboost within Fitted CHD | $\mu16 - \mu18$ |
| 18 | RF vs. Xgboost within Selected LiDAR | $\mu22 - \mu24$ |

**Table A5.** Ranking of LiDAR metrics based on RFE.

| Feature | Rank | Feature | Rank |
|---|---|---|---|
| elev_mean | 1 | elev_per50 | 9 |
| elev_max | 2 | elev_AIH10 | 10 |
| den_5 | 3 | elev_std | 11 |
| elev_per99 | 4 | elev_AIH80 | 12 |
| elev_AIH50 | 5 | den_9 | 13 |
| LAI | 6 | int_median | 14 |
| int_AIH90 | 7 | elev_min | 15 |
| int_max | 8 | int_per_10 | 16 |

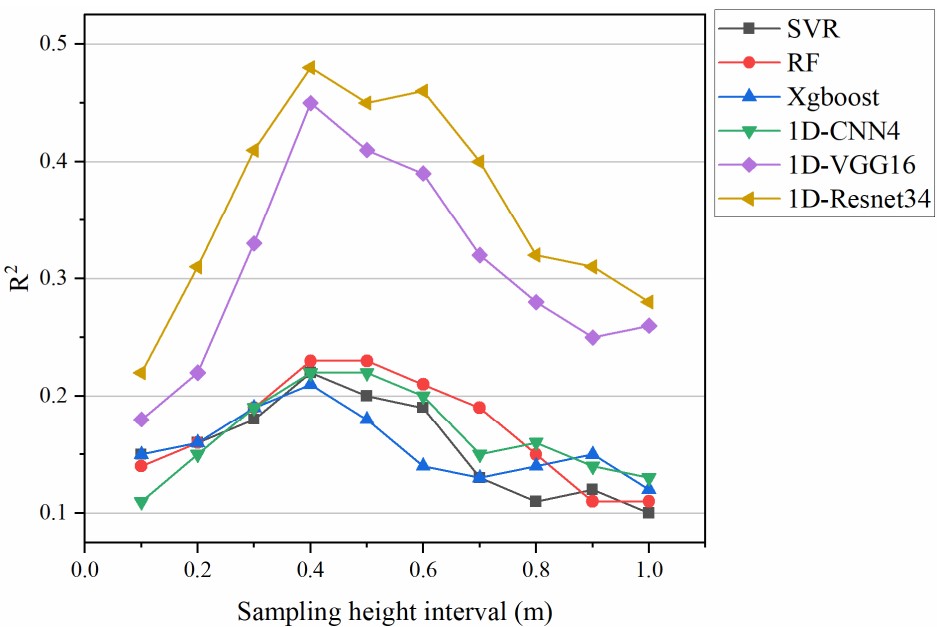

**Figure A1.** The effect of different sampling height intervals on model prediction accuracy (i.e., average $R^2$) of the AGB estimation for the six models (SVR, RF, Xgboost, 1D-CNN4, 1D-VGG16, and 1D-Resnet34).

**Table A6.** Performance of CHD in larch plantation forests.

| Feature (Feature Number) | Model | $R^2$ | RMSE (Mg/ha) | rRMSE |
|---|---|---|---|---|
| Original CHD | SVR | 0.28 | 27.33 | 0.25 |
| | RF | 0.27 | 29.15 | 0.27 |
| | Xgboost | 0.29 | 28.64 | 0.26 |
| Fitted CHD | SVR | 0.46 | 19.70 | 0.19 |
| | RF | 0.58 | 16.12 | 0.15 |
| | Xgboost | 0.57 | 15.09 | 0.16 |
| Original CHD | 1D-CNN4 | 0.26 | 21.84 | 0.22 |
| | 1D-VGG16 | 0.50 | 22.61 | 0.21 |
| | 1D-Resnet34 | 0.63 | 18.20 | 0.18 |
| Fitted CHD | 1D-CNN4 | 0.58 | 16.43 | 0.18 |
| | 1D-VGG16 | 0.72 | 14.89 | 0.16 |
| | 1D-Resnet34 | 0.88 | 8.27 | 0.08 |

Note: larch plantation sample plots were surveyed in the summer of 2020 at the Maoershan Experimental Forest Farm. A total of 30 sample plots (30 × 30 m) were surveyed and overlaid with ULS data.

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
