# Peer review of "Novel Features of Canopy Height Distribution for Aboveground Biomass Estimation Using Machine Learning: A Case Study in Natural Secondary Forests"

_remotesensing, doi:10.3390/rs15184364_

Round 1
Reviewer 1 Report
This manuscript proposed a novel feature of canopy height distribution which was fitting using a bimodal Gaussian function from UAV LiDAR data for aboveground biomass (AGB). In addition, the authors also provide a completely randomized design to investigate the effects of four feature sets and models. It can serve as a reference for selecting features and models to effectively estimate AGB. Overall, the manuscript is well written and provides reasonable analysis and conclusion. However, I still have some major concerns about the manuscript.
First, if the 62 plots in the study area are only distributed in Figure (c), why are not all the plots in (b) used for the analysis?
Second, what is the topography and slope of the 62 sample plots? Consider that topography affects the distribution of CHD curves. In addition, it can be seen in Fig. 6(b) that there are few ground points, and when the terrain is complex, it has a great impact on the CHD curve. It is suggested that the choice of models and features for different terrain conditions be analyzed and included in the Discussion section.
Third, add a table introducing the situation of the plots in Chapter 2.2.1.
Finally, the 62 plots used in this study are all mixed conifer-deciduous forests. Is your method universal in different forest types?

Recommendations to further improve the quality of English language.
Author Response
We appreciate the encouragement, careful review of the manuscript, and constructive comments and suggestions of all reviewers. After carefully considering all the comments, we have addressed every point of reviewers and made a number of revisions. All the changes are marked up using the “Track Changes” function in the text and listed below(attached file). The English of the entire manuscript is rechecked and polished.

Reviewer 2 Report
See attached review report

Author Response
We appreciate the encouragement, careful review of the manuscript, and constructive comments and suggestions of all reviewers. After carefully considering all the comments, we have addressed every point of reviewers and made a number of revisions. All the changes are marked up using the “Track Changes” function in the text and also listed below (attached file). The English of the entire manuscript is rechecked and polished.

Reviewer 3 Report
This article is very interesting with its scientific information and its method of presentation.
Just, two remarks were relieved
1-The title of publication is very complex, it must be changed as the following or as you like:
Using machine and deep learning in aboveground biomass estimation of secondary forest basing on Lidar data.
- 2 Figure 3 is not clearly readable, increase the font size
Author Response
Response: We appreciate the encouragement, careful review of the manuscript, and constructive comments and suggestions of all reviewers. After carefully considering all the comments, we have addressed every point of reviewers and made a number of revisions. All the changes are marked up using the “Track Changes” function in the text and also listed below.
- The title of publication is very complex, it must be changed as the following or as you like:
Using machine and deep learning in aboveground biomass estimation of secondary forest basing on Lidar data.
Response: Thank you for your comments. In order to highlight the proposed CHD features, we have shortened the title to the following:
“Novel Features of Canopy Height Distribution for Aboveground Biomass Estimation using Machine Learning: a case study in natural secondary forests”
- Figure 3 is not clearly readable, increase the font size
Response: Thank you for your comments. We have increased the font size and image size as following to make them clear for the reader.

Round 2
Reviewer 1 Report
The manuscript has been improved and the authors have addressed my concerns regarding the paper.